# Noncoplanar magnetic orders and gapless chiral spin liquid on the Kagome lattice with staggered scalar spin chirality

Fabrizio Oliviero[1⋆], João Augusto Sobral[2], Eric C. Andrade[2] and Rodrigo G. Pereira[1,3]

**1** Departamento de Física Teórica e Experimental,
Universidade Federal do Rio Grande do Norte, 59072-970 Natal, RN, Brazil
**2** Instituto de Física de São Carlos,
Universidade de São Paulo, C.P. 369, 13560-970, São Carlos, SP, Brazil
**3** International Institute of Physics, Universidade
Federal do Rio Grande do Norte, 59078-400 Natal, RN, Brazil

⋆ fabrizio.oliviero.108@ufrn.edu.br

## Abstract

Chiral three-spin interactions can suppress long-range magnetic order and stabilize quantum spin liquid states in frustrated lattices. We study a spin-1/2 model on the kagome lattice involving a staggered three-spin interaction $J_\chi$ in addition to Heisenberg exchange couplings $J_1$ on nearest-neighbor bonds and $J_d$ across the diagonals of the hexagons. We explore the phase diagram using a combination of a classical approach, parton mean-field theory, and variational Monte Carlo methods. We obtain a variety of noncoplanar magnetic orders, including a phase that interpolates between cuboc-1 and cuboc-2 states. In the regime of dominant $J_\chi$, we find a classically disordered region and argue that it may harbor a gapless chiral spin liquid with a spinon Fermi surface. Our results show that the competition between the staggered three-spin interaction and Heisenberg exchange interactions gives rise to unusual ground states of spin systems.



# 1  Introduction

In quantum magnets, the combination of competing exchange interactions and geometric frustration can lead to unconventional magnetic states [1]. One example is the emergence of noncoplanar spin structures in the Heisenberg model on the kagome lattice with further-neighbor interactions [2]. Besides the spontaneous magnetization, noncoplanar phases are distinguished by a scalar spin chirality [3]. In the extreme case, quantum fluctuations can melt the long-range magnetic order, giving rise to quantum spin liquids (QSLs) [4–6]. In chiral spin liquids (CSLs) [7–13], the ground state preserves the spin-rotation symmetry of the Hamiltonian, but supports a finite scalar spin chirality if reflection and time reversal symmetries are spontaneously or explicitly broken.

Chiral three-spin interactions provide a route to stabilizing CSL ground states [14–20]. Microscopically, this type of interaction arises in Mott insulators with a magnetic flux through triangular plaquettes, and their ratio to exchange interactions can be enhanced in the vicinity of the Mott transition [14,21]. In principle, the regime of strong three-spin interactions could be reached by Floquet engineering with circularly polarized light [22–24]. On the kagome lattice, a model with dominant three-spin interactions driving a uniform scalar spin chirality harbors the Kalmeyer-Laughlin CSL [10,14], a gapped topological phase with anyonic excitations. On the other hand, three-spin interactions that induce a staggered scalar spin chirality favor gapless CSLs [25–27], which are striking examples of non-Fermi liquids with Fermi surfaces of fractionalized excitations.

In this work we investigate a spin-1/2 model on the kagome lattice which includes staggered three-spin interaction $J_\chi$ as well as frustrated Heisenberg exchange interactions. Specifically, we consider exchange couplings $J_1$ on nearest-neighbor bonds and $J_d$ on bonds across the diagonals of the hexagons. Our motivation for studying this model is that an antiferromagnetic $J_d$ is the dominant interaction in the spin model for kapellasite [28–32], with ferromagnetic $J_1$ as the next-leading interaction. In this case, the system is expected to develop a noncoplanar magnetic order known as the cuboc-2 state [32,33]. On the other hand, based on numerical results and analytical arguments, a gapless CSL with a line Fermi surface has been proposed as the ground state of the model with chiral interactions only [27]. An important question pertains to the stability of this CSL with respect to magnetic order. Equivalently, from the perspective of the magnetic phases, one may ask how the chiral interaction destabilizes the cuboc order for a sufficiently large $J_\chi$. Besides addressing the stability of the gapless CSL within our numerical methods, our goal is to investigate the magnetic phases that appear when the exchange couplings and the chiral three-spin interaction become of the same order.

We start by mapping out the classical phase diagram of the model. Coming from the limit of dominant $J_d > 0$, we find that varying $J_1$ in the presence of a finite $J_\chi$ leads to a continuous transition to a noncoplanar phase that smoothly interpolates between the cuboc-2 and cuboc-1 states. This intermediate phase, which we call the AFMd phase, contains a variant of the octahedral state [2] as a special point at which the staggered spin chirality in the triangles of the kagome lattice is maximized. In our case, we observe antiferromagnetic chains in the diagonals of the hexagons of the kagome lattice, instead of the ferromagnetic chains in the original octahedral state, but the positions of the Bragg peaks in the spin structure factor are

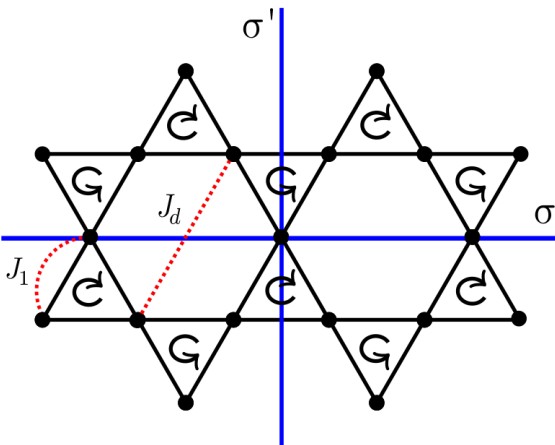

Figure 1: Schematic representation of the kagome model with Heisenberg exchange interactions $J_1$ and $J_d$ and chiral three-spin interaction $J_\chi$. The staggered chirality is represented by the arrows inside the triangles. The reflection axes $\sigma$ and $\sigma'$ are indicated by blue solid lines.

the same. In the regime where $J_\chi$ dominates, we observe a classically disordered region, compatible with the appearance of a QSL phase. To test this idea, we work out a parton mean-field theory [5, 6] of a U(1) CSL with a line Fermi surface. In contrast with previous work which considered Majorana fermions [27], here we construct variational wave functions using a parton representation with Abrikosov fermions, employing an ansatz classified in Ref. [9]. We compute the energy of the trial wave function using variational Monte Carlo (VMC) methods and compare it with the energy of competing classical states. In addition, we test the stability of the proposed CSL against order-inducing perturbations within the VMC approach. We find that the gapless CSL persists in a sizeable region in the phase diagram around the pure-$J_\chi$ point studied in Ref. [27].

The paper is organized as follows. In section 2, we present the $J_1$-$J_d$-$J_\chi$ model on the kagome lattice. In section 3, we explore the classical phase diagram and find novel ordered phases for both signs of $J_d$. Section 4 is devoted to the parton mean-field ansatz and the analysis of the spinon spectrum. In section 5, we show our VMC results and the phase diagram obtained by analyzing perturbations to the CSL variational wave function. We summarize our results in section 6. Finally, appendix A has some considerations about the nature of the classical degeneracy for the $J_1$-$J_d$-$J_\chi$ model, whereas appendix B contains some details of the derivation of the mean-field Hamiltonian for the three-spin interaction.

## 2 Model and symmetries

We consider an SU(2)-symmetric spin-1/2 model on the kagome lattice described by the Hamiltonian

$$H = H_0 + H_\chi, \tag{1}$$

where

$$H_0 = \sum_{ij} J_{ij} \mathbf{S}_i \cdot \mathbf{S}_j, \tag{2}$$

$$H_\chi = J_\chi \sum_{ijk \in \triangle} \mathbf{S}_i \cdot \left(\mathbf{S}_j \times \mathbf{S}_k\right) - J_\chi \sum_{ijk \in \nabla} \mathbf{S}_i \cdot \left(\mathbf{S}_j \times \mathbf{S}_k\right). \tag{3}$$

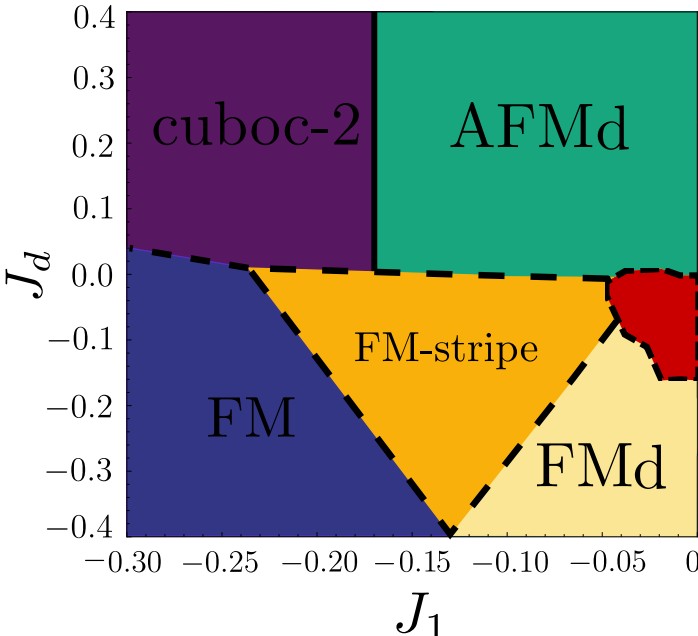

Figure 2: Classical phase diagram for the model in Eq. (1) on the kagome lattice with $J_\chi = 1$ setting the energy scale. We have five semiclassical ordered states: cuboc-2, AFMd, FM (ferromagnetic), FM-stripe, and FMd. Black dashed lines indicate first-order phase transitions. The solid line between the cuboc-2 and the AFMd phases indicates a continuous transition. A classically disordered region, in red, is present around the point $J_1 = J_d = 0$.

The nonzero exchange couplings in $H_0$ are $J_{ij} = J_1$ for nearest-neighbor bonds and $J_{ij} = J_d$ for bonds across the diagonals of the hexagons, see Fig. 1. In Eq. (3), the sites $i, j, k$ belong to an up-pointing ($\triangle$) or down-pointing ($\triangledown$) triangle and are oriented counterclockwise. The relative minus sign between the two terms in Eq. (3) induces a staggered scalar spin chirality. Without loss of generality, hereafter we set $J_\chi > 0$.

Besides breaking time-reversal symmetry, the chiral three-spin interaction lowers the point group symmetry of the Hamiltonian in comparison with the Heisenberg model on the kagome lattice. The rotational symmetry around the centers of the hexagons is reduced from sixfold to threefold. In addition, we can define reflections about two independent axes, indicated by $\sigma$ and $\sigma'$ in Fig. 1. The staggered chirality pattern breaks the reflection symmetry generated by $\sigma'$, but preserves $\sigma$.

Let us highlight two important limits of the model. For $J_d > 0$ and $J_d \gg J_\chi, |J_1|$, the Hamiltonian describes three sets of weakly coupled antiferromagnetic spin-1/2 chains rotated by 120° with respect to each other [26,32]. The low-energy physics of critical spin-1/2 chains is described by the SU(2)$_1$ Wess-Zumino-Witten (WZW) model [34]. However, the fixed-point of decoupled spin chains, $J_1 = J_\chi = 0$, is unstable against weak interchain couplings, and an arbitrarily small $J_1 < 0$ drives the system to the cuboc-2 phase [32]. In the limit $J_d = J_1 = 0$ and $J_\chi > 0$, there is compelling numerical evidence [25,27] that the ground state corresponds to a gapless CSL with a line Fermi surface protected by reflection symmetry $\sigma$ [9]. A signature of this gapless CSL is that spin correlations decay with distance $r$ as a power law $\sim r^{-2}$ in the directions perpendicular to the Fermi surface lines [26].

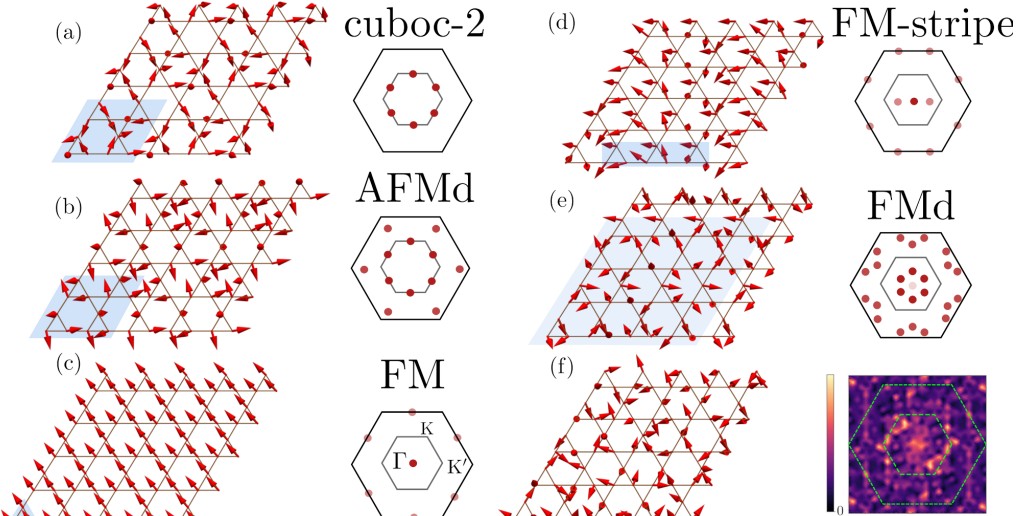

Figure 3: Classical spin configurations (left) and corresponding structure factor (right) for the phases in Fig. 2 with $J_\chi = 1$. The magnetic unit cell is marked by the blue shaded region. (a) cuboc-2; (b) AFMd ($J_1 = -0.05$, $J_d = 0.3$); (c) FM; (d) FM-stripe ($J_1 = -0.13$, $J_d = -0.16$); (e) FMd ($J_1 = -0.02$, $J_d = -0.3$); (f) classically disordered region. The color scale on the right is arbitrary. All snapshots show a small portion of a $L = 12$ lattice. The inner (outer) hexagon in the structure factor represents the original (extended) Brillouin Zone of the kagome lattice. In addition, the darker (lighter) the dot, the stronger (weaker) the relative intensity of the Bragg peaks located at this position. The FM-stripe is stabilized in one of three equivalent configurations distinguished by a $2\pi/3$ rotation. The state shown in (f) is only one of the many possible states inside the disordered region.

## 3 Classical phase diagram

To study the ordered phases for the model in Eq. (1), we start from the classical limit, and treat the spins as classical vectors of size $S$. Our main goal is to identify novel phases stabilized by the chiral interaction $J_\chi$. Because the chiral term contains a three-spin interaction, we cannot employ the usual Luttinger-Tisza method [35]. Instead, we numerically minimize Eq. (1) using a gradient descent algorithm. Given a spin $\mathbf{S}_i$, we anti-align it with respect to the gradient of the Hamiltonian: $\mathbf{S}_i^{m+1} = (1-\gamma)\mathbf{S}_i^m - \gamma\nabla_i\mathcal{H}(\mathbf{S}_i^m)$, with the step size $0 \leq \gamma \leq 1$ and $\nabla_i\mathcal{H} = \partial\mathcal{H}/\partial\mathbf{S}_i$ [36]. We consider $N_{\mathrm{cf}} \in [100, 200]$ distinct initial random spin configurations, and we sweep over the lattice locally minimizing each spin. We stop the algorithm when the overall change in the spin configuration after the $m$-th iteration is smaller than a given tolerance, which we typically set to $10^{-10}$. Our ground state is given by the spin configuration with the lowest energy in the final set. This procedure is realized on a kagome lattice with periodic boundary conditions and system size $N = 3 \times L \times L$ (see Sec. 4 for further details of the direct and reciprocal lattices). Specifically, we consider $L \in [6, 20]$ to investigate possible ordered phases with distinct magnetic unit cells. For a given classical ground state spin configuration, we compute its Fourier transform $\mathbf{S}_\mathbf{k} = N^{-1/2}\sum_j e^{-i\mathbf{k}\cdot\mathbf{r}_j}\mathbf{S}_j$, where $\mathbf{r}_j$ is the position of site $j$ and $\mathbf{k}$ is a wave vector. The static spin structure factor is given by $\mathcal{S}(\mathbf{k}) = \mathbf{S}_\mathbf{k} \cdot \mathbf{S}_{-\mathbf{k}} = |\mathbf{S}_\mathbf{k}|^2$. The order parameter is then given by $m^2 = \mathcal{S}(\mathbf{Q})/N$, where $\mathbf{Q}$ is the ordering wave vector, corresponding to the location of the Bragg peaks in $\mathcal{S}(\mathbf{k})$.

Our procedure works as follows. For a fixed set $(J_1, J_d, J_\chi)$, we find the ground state spin configuration and its structure factor for a given system size $N$. We vary the parameters

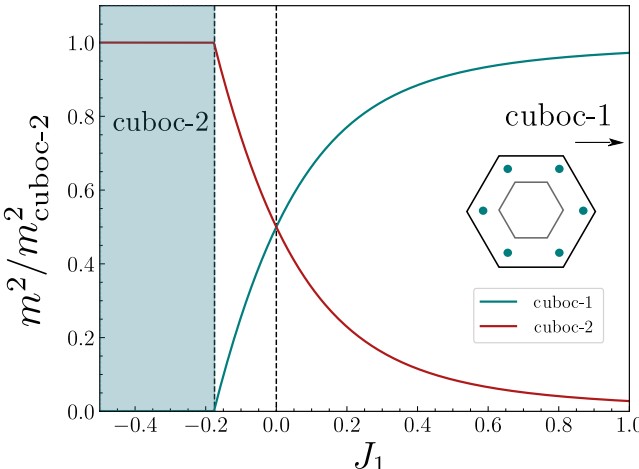

Figure 4: Order parameter squared (normalized with respect to its value in the cuboc-2 state) as a function of $J_1$, indicating the continuous transition between the cuboc-2 and AFMd phases for $J_\chi = 1$ and $J_d = 0.2$. This is true for all $J_d > 0$ where the cuboc phases are stable. The red (light-blue) curve shows the normalized Bragg peak intensity at the cuboc-2(1) ordering wave vectors. At the point $J_1 = 0$, an octahedral state maximizing the staggered chirality emerges. The phase transition to the cuboc-1 is slower and takes place in the vicinity of $J_1 = 1$ (not shown). Inset: structure factor for the cuboc-1 phase.

identifying phase transitions by sudden changes in $\mathcal{S}(\mathbf{k})$ and peaks in $-\partial^2 E_0/\partial J_{1,d}^2$, where $E_0$ is the classical ground state energy. We repeat this procedure for different system sizes to accommodate commensurate spiral orders. Figure 2 shows the classical phase diagram obtained within this formalism. In addition to the previously reported cuboc-2 [Fig. 3(a)] and FM phases [Fig. 3(c)] [2, 37], we find three other magnetically ordered phases: AFMd [Fig. 3(b)], FM-stripe [Fig. 3(d)], and FMd [Fig. 3(e)]. These three phases display a net nonzero staggered chirality on the corner-sharing triangles of the kagome lattice, indicating that they are stabilized by $J_\chi$. We encounter no incommensurate spiral orders, but uncover the existence of an extended classically disordered region for dominant $J_\chi$, where no sign of magnetic order is detected, Fig. 3(f).

The AFMd state, Fig. 3(b), is characterized by three antiferromagnetic spin chains running along the diagonals of the hexagons in the kagome lattice, with the relative orientation among the chains controlled by $J_1$ and $J_\chi$. For fixed $J_\chi$, the AFMd phase interpolates between the cuboc-1 and cuboc-2 phases as we vary $J_1$, Fig. 4. Because these three phases display the same magnetic unit cell, by varying $J_1$ we smoothly modify the relative intensity of the Bragg peaks, leading to a continuous phase transition as the peaks corresponding to one of the cuboc phases vanish. For $J_1 = 0$, all Bragg peaks have the same intensity. This point corresponds to a variant of the octahedral state [2], for which the classical scalar staggered spin chirality is maximized.[1]

A large and negative $J_1$ favors the FM state with fully polarized spins, Fig. 3(c). As we reduce the absolute value of $J_1$, a new phase appears, the FM-stripe, Fig. 3(d). In this state, ferromagnetic spin chains appear along a single diagonal in the hexagons, with the spins in the other two diagonals displaying a small angle between them. Importantly, this phase possesses a finite spin chirality in the triangles, indicating an energetic trade-off between $J_1$ and $J_\chi$. By symmetry, there are two other equivalent spin configurations, with the ferromagnetic chains

---

[1]The continuous evolution of the AFMd phase in real space can be found in the supplemental material at https://github.com/joaosds/suppl_noncoplanar for fixed $J_\chi = 1$ and $J_d > 0$ while $J_1$ is varied.

running along one of the other two diagonals. As in the AFMd case, the relative intensity of the Bragg peaks varies with $J_1$.

If one starts from the limit $J_1 = 0$, a negative $J_d$ favors ferromagnetic chains along the diagonals of the hexagons. This gives rise to the FMd phase, Fig. 3(e), in analogy to the AFMd. The magnetic unit cell, however, contains 48 spins as opposed to the 12 spins in the AFMd, and the relative intensity of the Bragg peaks also depends on $J_1$. We find that the transitions between the FM and FM-stripe and FM-stripe and FMd are discontinuous. We leave a more detailed characterization of the magnetically ordered phases for future work.

Finally, we address the classically disordered region. In all its extent, the static spin structure factor shows neither Bragg peaks nor sharp features, and its weight is distributed over the entire Brillouin zone, Fig. 3(f). A classically disordered region is tied to the presence of massively degenerate states and usually occurs at isolated points in the phase diagram, for instance at the boundaries between two ordered phases. Its extended nature in the present problem may be traced back to the frustrating nature of the kagome lattice. In fact, the classical model at the pure chiral point with $J_1 = J_d = 0$ is known to have an extensive ground state degeneracy which is not completely lifted by $J_1 \neq 0$ [38]. We elaborate on this point in appendix A. Although quantum fluctuations may lift the massive degeneracy via the order-by-disorder mechanism [38–41], the presence of an extended classically disordered region in the regime $J_\chi \gg |J_1|, |J_d|$ is a promising sign that a CSL might be stable for $S = 1/2$, as indicated by numerical results for the pure chiral model [27].

# 4 Parton mean-field theory for gapless chiral spin liquid

As discussed in Sec. 3, the classical phase diagram features a disordered region that may support a QSL ground state for $S = 1/2$. To describe this state, we employ a parton construction in which we fractionalize the spin operator into fermionic spinons, also called Abrikosov fermions [6, 42].

Our choice of fermionic spinons is motivated by the suggestion of a gapless CSL for the model with staggered scalar spin chirality [25–27]. The projective symmetry group classification of U(1) chiral spin liquids with fermionic partons on the kagome lattice can be found in Ref. [9]. In the following, we focus on a specific ansatz that is compatible with all symmetries of the Hamiltonian and reproduces a line Fermi surface in the spinon spectrum, as expected for the model with $J_1 = J_d = 0$. This choice can be justified a posteriori since we will show that it generates a competitive variational wave function, whose energy is lower than that of the magnetically ordered states. Since in the following we restrict the number of mean-field parameters and do not explore all possible Ansätze, we cannot rule out the possibility of a better ansatz which yields an even lower energy.

We introduce charge-neutral spin-1/2 fermions $f_{i\alpha}$, with $\alpha = \uparrow, \downarrow$, which satisfy the algebra $\{f_{i\alpha}, f_{j\beta}^\dagger\} = \delta_{ij}\delta_{\alpha\beta}, \{f_{i\alpha}, f_{j\beta}\} = 0$. The spin operator at site $i$ is written as

$$S_i^a = \frac{1}{2}\sum_{\alpha\beta} f_{i\alpha}^\dagger (\sigma^a)_{\alpha\beta} f_{i\beta}, \tag{4}$$

where $\sigma^a$ are Pauli matrices. Following the standard parton mean-field decoupling of the Heisenberg interactions in Eq. (1) [42], we obtain

$$H_0^{\mathrm{MF}} = -\sum_{\alpha, ij} \frac{J_{ij}}{2}\left(\xi_{ji} f_{i\alpha}^\dagger f_{j\alpha} + h.c.\right) + \sum_{ij} \frac{J_{ij}}{2}\left|\xi_{ij}\right|^2, \tag{5}$$

with $\xi_{ij} = \sum_\alpha \langle f_{i\alpha}^\dagger f_{j\alpha}\rangle$ a mean-field parameter that specifies the QSL ansatz. This description leads to a U(1) gauge redundancy [43]. In order to recover physical states, we must impose

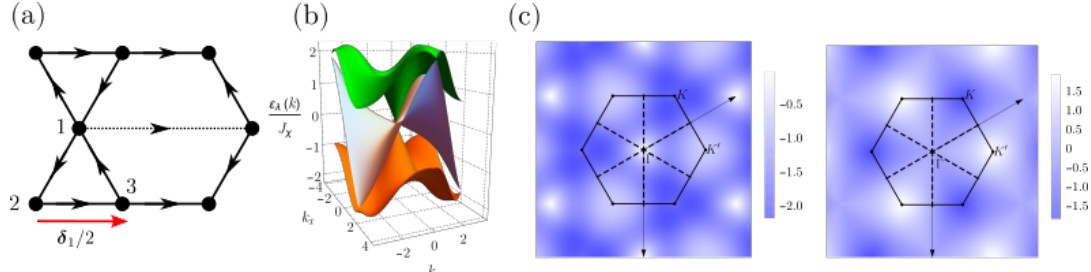

Figure 5: (a) Mean-field ansatz on the kagome lattice. Sites are labeled by the corresponding sublattice index $s \in \{1,2,3\}$, see Eq. (9). The arrows on the solid lines indicate the bond direction in which $\xi_{ij} = i\xi_1$. Likewise, $\xi_{ij} = i\xi_d$ in the direction indicated by the arrow on the dashed line. The red arrow represents the nearest-neighbor vector $\boldsymbol{\delta}_1/2$. (b) Three-dimensional plot of the spinon dispersion for $\kappa_d/\kappa_1 = 1.2$. (c) Density plot of the dispersion relations for the lower band $\varepsilon_1(k)$ (left) and the middle band $\varepsilon_2(k)$ (right). The dashed lines represent the spinon Fermi surface for the middle band.

the single-occupancy constraint locally

$$\sum_\alpha n_{i\alpha} = \sum_\alpha f_{i\alpha}^\dagger f_{i\alpha} = 1, \quad \forall i. \tag{6}$$

The three-spin interaction in Eq. (2) can also be decoupled into fermion bilinears using the same mean-field parameters [44]. For instance, the contribution from the up-pointing triangles takes the form (see appendix B)

$$H_\chi^{\mathrm{MF}} = \frac{3iJ_\chi}{16} \sum_{ijk\in\triangle} \sum_\alpha \left[ -\xi_{ik}\xi_{kj}\xi_{ji} + \xi_{kj}\xi_{ji}f_{i\alpha}^\dagger f_{k\alpha} + \xi_{ik}\xi_{kj}f_{j\alpha}^\dagger f_{i\alpha} + \xi_{ji}\xi_{ik}f_{k\alpha}^\dagger f_{j\alpha} - \text{h.c.} \right]. \tag{7}$$

On the kagome lattice, we denote by $f_{s\alpha}(\mathbf{R})$ the annihilation operator for a fermion with spin $\alpha$ on sublattice $s \in \{1,2,3\}$ of the unit cell at position $\mathbf{R}$. As our ansatz, we consider a U(1) CSL given by a staggered flux phase classified as No. 11 in Table IX of Ref. [9], where we set the second-neighbor exchange coupling $J_2 = 0$. The unit cell is defined as an up-pointing triangle, see Fig. 5(a). The lattice vectors are $\boldsymbol{\delta}_1 = (1,0), \boldsymbol{\delta}_2 = (-1/2, \sqrt{3}/2)$, and $\boldsymbol{\delta}_3 = (-1/2, -\sqrt{3}/2)$. Using translational invariance, we introduce the notation

$$\xi(s, s'; \mathbf{v}) = \sum_\alpha \langle f_{s\alpha}^\dagger(\mathbf{R} + \mathbf{v}) f_{s'\alpha}(\mathbf{R}) \rangle. \tag{8}$$

The mean-field amplitudes are taken as imaginary numbers,

$$\xi(s+1, s; \mathbf{0}) = -\xi(s+1, s; -\boldsymbol{\delta}_{s-1}) = i\xi_1, \tag{9}$$

$$\xi(s, s; \boldsymbol{\delta}_s) = -i\xi_d, \tag{10}$$

so that $\xi(s, s'; \mathbf{v}) = \xi^*(s', s; -\mathbf{v})$ with $s + 3 \equiv s$. Here $\xi_1$ and $\xi_d$ are real order parameters. The gauge flux $\Phi_\triangle$ through an up-pointing triangle is defined by $\xi(3, 2; \mathbf{0})\xi(2, 1; \mathbf{0})\xi(1, 3; \mathbf{0})$ $= -\xi_1^3 = |\xi_1|^3 e^{i\Phi_\triangle}$. Thus, $\Phi_\triangle = -\frac{\pi}{2}\text{sgn}(\xi_1)$. The three-spin interaction with $J_\chi > 0$ selects negative chirality on up-pointing triangles, which corresponds to $\xi_1 > 0$. We can also characterize the ansatz by the flux through a trapezoid with the longer base along a $J_d$ bond, see Fig. 5(a). We obtain zero flux if $\xi_1$ and $\xi_d$ have the same sign and $\pi$ flux otherwise.

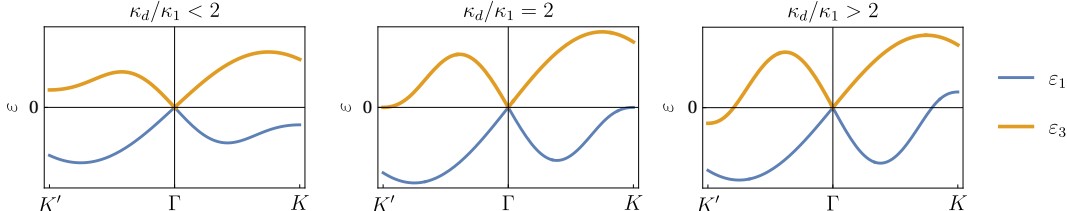

Figure 6: Spinon dispersion for the lower and upper bands, $\varepsilon_1(\mathbf{k})$ and $\varepsilon_3(\mathbf{k})$, along the $\Gamma$-K direction, for different values of the ratio $\kappa_d/\kappa_1$. For $\kappa_d/\kappa_1 < 2$, the bands are gapped at the K and K$'$ points. At the critical value $\kappa_d/\kappa_1 = 2$, the bands cross the Fermi level at the K and K$'$ points, forming Fermi pockets for $\kappa_d/\kappa_1 > 2$.

We can diagonalize the mean-field Hamiltonian by taking the Fourier transform of the fermion operators:

$$f_{s\alpha}(\mathbf{R}) = \sqrt{\frac{3}{N}} \sum_{\mathbf{k}} e^{i\mathbf{k}\cdot(\mathbf{R}+\mathbf{a}_s)} f_{s\alpha}(\mathbf{k}), \tag{11}$$

where $\mathbf{a}_1 = 0$, $\mathbf{a}_2 = \boldsymbol{\delta}_3/2$ and $\mathbf{a}_3 = -\boldsymbol{\delta}_2/2$ are the relative positions within the unit cell and $N$ is the total number of sites. The mean-field Hamiltonian takes the form

$$H_{\text{MF}} = \sum_{\alpha}\sum_{\mathbf{k}} \psi_{\mathbf{k}\alpha}^{\dagger} \mathcal{H}(\mathbf{k}) \psi_{\mathbf{k}\alpha} + NJ_1\xi_1^2 + \frac{NJ_d\xi_d^2}{2} + \frac{NJ_\chi\xi_1^3}{2}, \tag{12}$$

with the spinor $\psi_{\mathbf{k}\alpha} = (f_{1\alpha}(\mathbf{k}), f_{2\alpha}(\mathbf{k}), f_{3\alpha}(\mathbf{k}))^T$. The Bloch Hamiltonian $\mathcal{H}(\mathbf{k})$ is given by

$$\mathcal{H}(\mathbf{k}) = -\begin{pmatrix} \kappa_d\sin(k_1) & \kappa_1\sin(k_3/2) & \kappa_1\sin(k_2/2) \\ \kappa_1\sin(k_3/2) & \kappa_d\sin(k_2) & \kappa_1\sin(k_1/2) \\ \kappa_1\sin(k_2/2) & \kappa_1\sin(k_1/2) & \kappa_d\sin(k_3) \end{pmatrix}, \tag{13}$$

where $k_i = \mathbf{k}\cdot\boldsymbol{\delta}_i$ and we define the effective hopping amplitudes $\kappa_1 = \frac{3}{8}J_\chi\xi_1^2 - J_1\xi_1$ and $\kappa_d = J_d\xi_d$. For $J_\chi > 0$ and $J_1 < 0$, we have $\kappa_1 > 0$. For $J_d > 0$, the sign of $\kappa_d$ depends on $\xi_d$, which is related to the gauge flux on trapezoids. Diagonalizing $\mathcal{H}(\mathbf{k})$, we obtain the dispersion relations $\varepsilon_\lambda(\mathbf{k})$, where $\lambda = 1, 2, 3$ is a band index. Due to particle-hole symmetry, $\varepsilon_\lambda(\mathbf{k}) = -\varepsilon_{4-\lambda}(-\mathbf{k})$, the chemical potential must be set to $\mu = 0$ to satisfy the half-filling condition $\sum_\alpha \langle f_{i\alpha}^\dagger f_{i\alpha} \rangle = 1$. The mean-field ground state $|\Psi_{\text{MF}}\rangle$ is identified with a Fermi sea in which all negative-energy single-particle states are occupied.

Figure 5(b) shows the spectrum for $\kappa_1 > 0$ and $\kappa_d/\kappa_1 = 1.2$, which is representative of the parameter regime $0 < \kappa_d < 2\kappa_1$. In this case, the lower and upper bands exhibit a Dirac cone at the $\Gamma$ point of the Brillouin zone. The middle band shows gapless lines along the $\Gamma$-M directions, see Fig. 5(c). The Fermi surface lines are robust against variations of the ratio $\kappa_d/\kappa_1$ and their location is fixed by the reflection symmetry $\sigma$. However, as we increase $\kappa_d/\kappa_1$, the energy gap for the lower and upper bands at the K and K$'$ points decreases. Precisely at the critical value $\kappa_d/\kappa_1 = 2$, these bands cross the Fermi level with a quadratic dispersion at the K and K$'$ points, as represented in Fig. 6. The crossing of the Fermi level and subsequent formation of Fermi pockets around K and K$'$ for $\kappa_d/\kappa_1 > 2$ signals a nesting instability [45] of our CSL at large $\kappa_d$. We have also looked at the spectrum for $\kappa_d < 0$. In this case, a possible instability of the CSL is indicated by a flattening of the middle band around $\kappa_d/\kappa_1 = -1$. For either sign of $\kappa_d$, the instabilities at large $|\kappa_d|$ can be associated with the regime of dominant $J_d$ interaction, where we expect the gapless CSL to be replaced by the ordered phases discussed in Sec. 3. This analysis shows that the range of mean-field parameters where we may consider a CSL wave function must be restricted to $-1 < \kappa_d/\kappa_1 < 2$.

# 5 Variational Monte Carlo results

With the CSL ansatz at hand, we can construct a free fermion wave function that is invariant under the symmetries of the system once the single-site occupancy, Eq. (6), is enforced. However, the ansatz tells us nothing about the energy of these wave functions. To obtain reliable energy estimates, we carry out a variational analysis based on the projection of the mean-field wave function in the region $-1 < \kappa_d/\kappa_1 < 2$. Specifically, we enforce the constraint in Eq. (6) considering a Gutzwiller projection

$$\hat{\mathcal{P}}_G = \prod_i \left( n_{i\uparrow} - n_{i\downarrow} \right)^2 . \tag{14}$$

To compare the energy of our CSL state to that of the ordered states discussed in Sec. 2, we rewrite Eq. (12) as

$$\tilde{\mathcal{H}}_{\text{MF}} = \sum_{\alpha,ij} \kappa_{ij} f_{i\alpha}^\dagger f_{j\alpha} + h \sum_i \mathbf{M}_i \cdot \mathbf{S}_i . \tag{15}$$

Besides the oriented hopping structure encoded in $\kappa_{ij}$, as discussed in Sec. 4, we include a Zeeman term. Here $h$ controls the strength of the Zeeman coupling, and $\mathbf{M}_i$ is a classical spin configuration corresponding to one of the ordered states in Fig. 2. It suffices to consider $h \geq 0$. The vector $\mathbf{M}_i$ effectively acts as a staggered magnetic field and magnetic order can be induced on top of the CSL state if $h \neq 0$ variationally [46,47]. In this situation, the spinon spectrum is gapped and we interpret the resulting state as adiabatically connected to a conventional magnetically ordered one.

We performed VMC simulations [48] and measured the ground state energy, $E = \langle \Psi | H | \Psi \rangle$, with $H$ given in Eq. (1) and

$$|\Psi\rangle = \hat{\mathcal{P}}_G |\tilde{\Psi}_{\text{MF}}\rangle . \tag{16}$$

Here, $|\tilde{\Psi}_{\text{MF}}\rangle$ is the ground state of Eq. (15) at half-filling, with the Gutzwiller projector $\hat{\mathcal{P}}_G$ given by Eq. (14). In general, including local correlations in our variational wave function – for instance adding Jastrow factors or optimizing the classical angles – will reduce the energy of the ordered states further. We refrain to do so here to limit the number of variational parameters and to explore in detail the phase diagram. Even with this simplification, we have a flexible ansatz containing energetically competitive magnetic states in addition to the CSL.

In the VMC simulations, we randomly place each spinon spin flavor on $N/2$ sites of our lattice. Our VMC moves consist of exchanging a random pair of sites of distinct spin flavors. The exchanges are accepted or rejected according to the Metropolis-Hastings algorithm [49]. The probability of each configuration is proportional to the square of the wave function. A number $N$ of exchanges attempts define a VMC sweep. After $N_{\text{warm}} \sim 10^4$ sweeps for thermalization, we calculate our observables considering further $N_{\text{meas}} \sim 10^4$ sweeps. We take $\kappa_d$ and $h$ as our variational parameters, setting $\kappa_1 = 1$ as an inconsequential energy scale. Besides the state discussed in Sec. 4, we also considered an ansatz with $\pi/2$-flux on the trapezoids. This extra case corresponds to ansatz No. 9 in Table IX of Ref. [9]. In contrast with the state discussed in Sec. 4, ansatz No. 9 allows for a more general shape of the Fermi surface. We find its energy not to be competitive and we refrain from discussing it further. We consider periodic boundary conditions for $H$ and work with systems sizes up to $L = 14$. To mitigate finite-size effects, we implement mixed boundary conditions for $\tilde{H}_{\text{MF}}$ [50,51]. Specifically, we consider periodic boundary conditions along the $\boldsymbol{\delta}_1$ direction and antiperiodic boundary conditions in the $\boldsymbol{\delta}_2$ direction.

The VMC result for the CSL limit ($h = 0$) is shown in Fig. 7 as a function of $\kappa_d$. For large $|\kappa_d|$, our ansatz recovers the energy of the antiferromagnetic Heisenberg chain in the limit $J_d \gg J_\chi, |J_1|$ [52]. As we discussed in Sec. 2, we expect the CSL to be unstable in this limit.

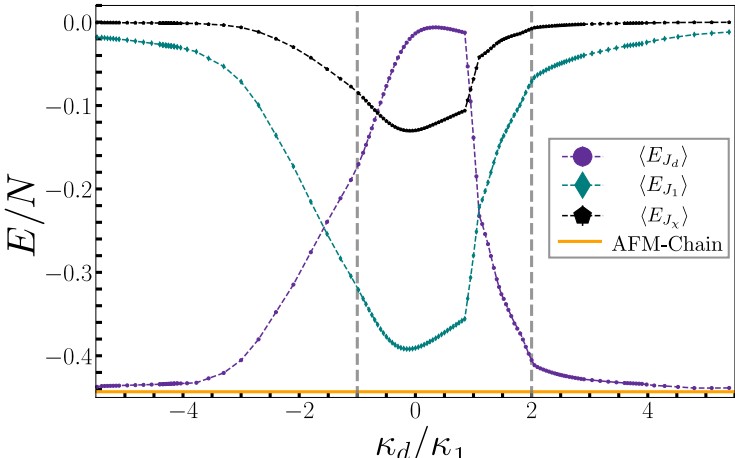

Figure 7: Variational energy $E = \langle \Psi | H | \Psi \rangle$ as a function of $\kappa_d$ for $h = 0$ obtained via VMC. We calculate the expectation values of all three terms in Eq. (1): chiral term (black curve), Heisenberg $J_1$ (green curve) and $J_d$ (purple curve) terms. For comparison, we show the exact ground state energy of the antiferromagnetic Heisenberg chain (orange line). The vertical dashed lines indicate the region of stability for the CSL ansatz as discussed in Sec. 4. We considered $L = 12$ and error bars are smaller than the symbol markers.

Inside the stability range of the CSL ansatz, $-1 < \kappa_d < 2$, we observe that the minimum value of the energy occurs for $\kappa_d \approx -0.1$. The energy, per spin, of the CSL state is then given by

$$E_{\text{CSL}}/N = -0.392(1)J_1 - 0.015(1)J_d - 0.131(1)J_\chi, \qquad (17)$$

in the limit $J_\chi \gg |J_d|, |J_1|$. An alternative competitive ansatz for the CSL comes from a parton construction in terms of Majorana fermions [25]. We find that the energy of this state is the same as the one in Eq. (17) as long as one does not include a BCS-like $p$-wave pairing in $|\tilde{\Psi}_{\text{MF}}\rangle$. For the sake of simplicity, we did not pursue this possibility.

We are now in position to explore the stability of the CSL with respect to the magnetically ordered phases present in Fig. 2. In our variational language, we say that a given ordered state is selected if the energy is minimized for $h \neq 0$. To complement the characterization of the ordered phases, we compute the square of the sublattice magnetization $m$

$$m^2 = \lim_{|i-j| \to \infty} \langle \mathbf{S}_i \cdot \mathbf{S}_j \rangle. \qquad (18)$$

This observable estimates the spin-spin correlation at the maximum distance for two spins belonging to the same magnetic sublattice and gives the square of the staggered magnetization. We then have $m = 0$ in the CSL and $m > 0$ in the corresponding magnetically ordered phase. In Fig. 8 we show the phase transition between the CSL and the AFMd phase as we vary $J_d$ for $J_1 = -0.01$. The finite size-scaling allows us to estimate the transition taking place at $J_d = 0.08(2)$, showing that the CSL is stable around the classically disordered region. Notice that in the ordered phase we have $m < S$ due to the quantum fluctuations captured by our variational calculation.

In practice, we construct our phase diagram mainly considering a fixed system size ($L = 12$) due to the complex nature of the ordered phases present. Since the classical spin configurations of the AFMd, FMd and FM-stripe phases depend on $J_1$, but not on $J_d$, we compare the energy of the different phases by fixing $J_1$ and varying $J_d$ in the VMC simulation. For a given $J_1$, we then extract the classical spin configuration $\mathbf{M}_i$. These spins act as a rigid staggered field on top of the spin-liquid phase. The sole variational parameter is $h$. The resulting

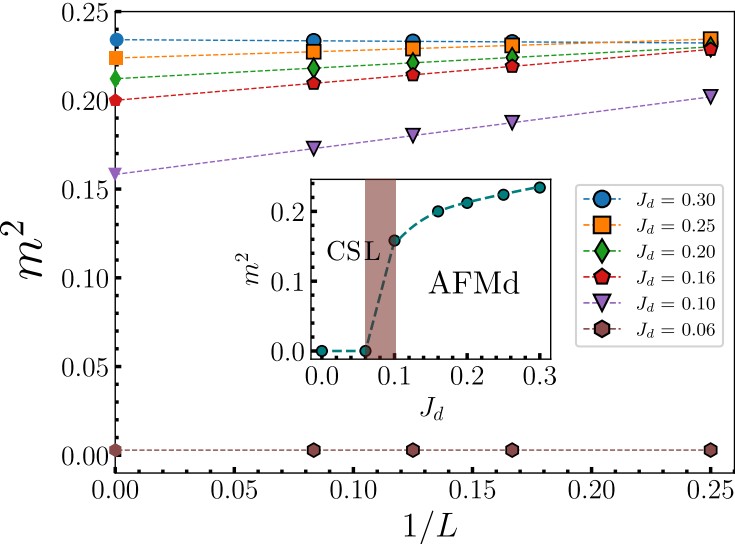

Figure 8: Finite-size scaling for the magnetic order parameter as a function of $1/L$ for $J_1 = -0.01$, $J_\chi = 1$, and several values of $J_d$. Inset: Extrapolation of $m^2$ to the thermodynamic limit. $m^2 = (>)0$ corresponds to the CSL (AFMd) phase. The colored region indicates the numerical uncertainty in the location of the phase transition.

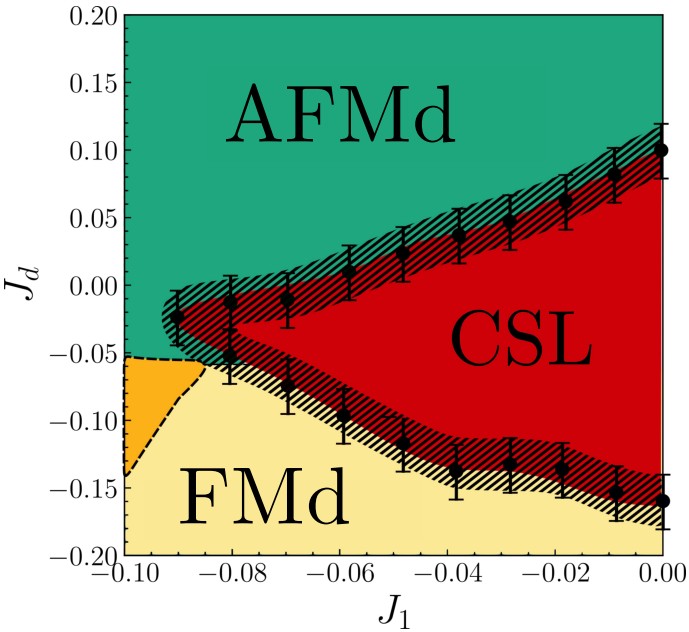

Figure 9: Phase diagram of the model in Eq. (1) on the kagome lattice obtained via VMC for $J_\chi = 1$ and $S = 1/2$. We focus on the region around the chiral spin liquid (CSL) phase. The classical representation of the magnetically ordered phases is shown in Fig. 3.

phase diagram for $S = 1/2$, in the vicinity of the CSL, is displayed in Fig. 9. The error bars in Fig. 9 come mainly from the presence of magnetization plateaus, due to finite size effects, which hamper a precise determination of the phase boundaries. Overall, the CSL is expanded with respect to the classically disordered region, towards both the AFMd and FM-stripe phases, strongly suggesting the selection of a CSL by a dominant $J_\chi$.

The position of the transition between the magnetically ordered phases is also altered. Quantum fluctuations reduce the width of the FM-stripe phase with respect to the FMd phase, in the vicinity of the CSL where $|J_d| > |J_1|$. The FMd phase has three coupled FM chains along the diagonal of the hexagons [Fig. 3(e)], as opposed to a single FM chain in the FM-stripe [Fig. 3(d)], which could explain its relative stability despite the larger magnetic unit cell.

## 6   Discussion

We investigated the rich phase diagram that stems from the competition between staggered three-spin interactions and frustrated Heisenberg interactions on the extended kagome lattice. In the regime of dominant three-spin interactions, our results support the existence of the gapless chiral spin liquid phase identified in Refs. [25, 27]. The classically disordered region that we observed in this regime is consistent with the analysis of trial wave functions by variational Monte Carlo, which shows that the chiral spin liquid state has lower energy and is stable against order-inducing perturbations.

Increasing the strength of the Heisenberg interactions, we found several noncoplanar magnetic states beyond the previously reported cuboc phases. The AFMd and FMd phases can be pictured as coupled spin chains with Néel or ferromagnetic order, respectively, running along the diagonals of the hexagons in the kagome lattice. The angle between the magnetization in different sets of crossing chains varies continuously with the nearest-neighbor exchange coupling, and the cuboc-1, cuboc-2, and octahedral states can be viewed as particular limits of the AFMd phase. We found a continuous transition from the AFMd phase to the cuboc-2 phase which is manifested in the relative intensity of Bragg peaks in the spin structure factor. This continuous transition is reminiscent of the transition from canted antiferromagnetism to the fully polarized state driven by an external magnetic field [53], but here it is driven by a compromise between the frustrating exchange couplings and the three-spin interaction. In addition, we identified an FM-stripe phase at intermediate couplings. This phase breaks the $C_3$ lattice rotational symmetry as the spins select one out of the three diagonals of the hexagons to form ferromagnetically ordered spin chains.

As an extension of this work, it would be interesting to further characterize the novel magnetic phases, in particular by investigating the effects of thermal and quantum fluctuations [38]. Another important question pertains to the nature of the quantum phase transitions from the gapless chiral spin liquid to the magnetically ordered phases. For the topological chiral spin liquid with uniform scalar spin chirality, numerical evidence indicates that exotic continuous transitions may take place as a result of quantum melting of the noncoplanar order [18]. In contrast, transitions from the gapless chiral spin liquid with staggered spin chirality remain largely unexplored.

## A   Classical ground state degeneracy near the chiral point

In section 3 we encountered an extended disordered region in the classical model with dominant three-spin interactions. Generically, a classically disordered phase can be connected to classically degenerate states and often appears at the boundaries between two ordered phases.

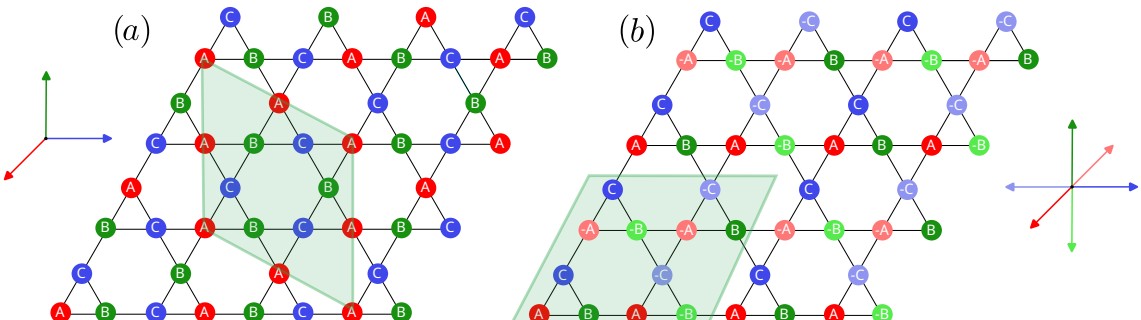

Figure 10: (a) Example of a three-color triaxial state. The directions A, B, and C form an orthogonal basis. (b) Variant of the octahedral phase with AFM chains along the diagonals of the hexagons. The axis are illustrative and not unique. The unit cell is represented by the parallelograms.

Its extended nature in the present problem may be traced back to the frustrated geometry of the kagome lattice combined with the effects of the three-spin interaction.

The extensive degeneracy of the classical $J_1$-$J_\chi$ model was discussed in Ref. [38] for $J_1 > 0$ and both uniform and staggered chirality. Let us take the pure chiral model, $J_1 = J_d = 0$, as our reference point. The energy of the staggered chiral interaction is minimized imposing that the spins in the up (down) triangles form a right (left)-handed orthogonal basis. An important subset of theses states are the triaxial states [38], shown in Fig. 10 (a), in which each spin is collinear with one of three directions represented by three colors. Triaxial states in the pure chiral model have an extensive degeneracy which scales as $2^{N/6}$ and is associated with local $\mathbb{Z}_2$ degrees of freedom. The latter is reminiscent of the degeneracy of coplanar states for the antiferromagnetic Heisenberg model on the kagome lattice [54]. In the presence of nonzero $J_1$, the triaxial states can be generalized by considering three directions which are no longer orthogonal. In terms of the angle $\theta$ that the spins form with the space diagonal, the energy for a single triangle is $\mathcal{E}(\theta) = \frac{3}{4}[J_1 S^2(1 + 3\cos 2\theta) - \sqrt{3} J_\chi S^3 \sin\theta \sin 2\theta]$. For $J_1 < 0$, the angle $\theta_0$ that minimizes the energy decreases with $|J_1|$ until we reach the critical value $J_{1c} = -S/\sqrt{3}$. For $J_1 < J_{1c}$, we obtain $\theta_0 = 0$, corresponding to the ferromagnetic state. On the other hand, for $J_1 > J_{1c}$ the classical ground state remains massively degenerate because one can construct a subextensive set of states which are degenerate with a given three-color state, as explained in Ref. [38].

The construction of classically degenerate ground states for the $J_1$-$J_\chi$ model does not hold once we add the exchange coupling $J_d$. In fact, starting from the pure chiral point, a small $J_d > 0$ has an immediate impact: it selects triaxial states with AFM chains along the diagonals of the hexagons, shown in Fig. 10(b). This state minimizes both the $J_\chi$ and $J_d$ terms. As a result, the extensive ground state degeneracy is lifted at first order in $J_d > 0$ and we enter the AFMd phase, see Fig. 2. The situation for $J_d < 0$ is distinct because FM chains running along the diagonals of the hexagons are incompatible with triaxial states. Within the set of triaxial states, those in which spins across the diagonals point in perpendicular directions, as in Fig. 10(a), have lower energy than the AFMd state in Fig. 10(b). However, the criterion of triaxial states with orthogonal spins across the diagonals still leaves an extensive residual degeneracy due to the $\mathbb{Z}_2$ degrees of freedom. On the other hand, in the states obtained by the gradient descent minimization algorithm for small $J_d < 0$, such as the one illustrated in Fig. 3(f), the spins within the same unit cell remain approximately orthogonal to each other, but the directions of the axes vary in an apparently disordered fashion between different unit cells. Thus, they are not obviously related to triaxial states. While we have not been able to identify the local transformations that may connect these ground states, our numerical results strongly

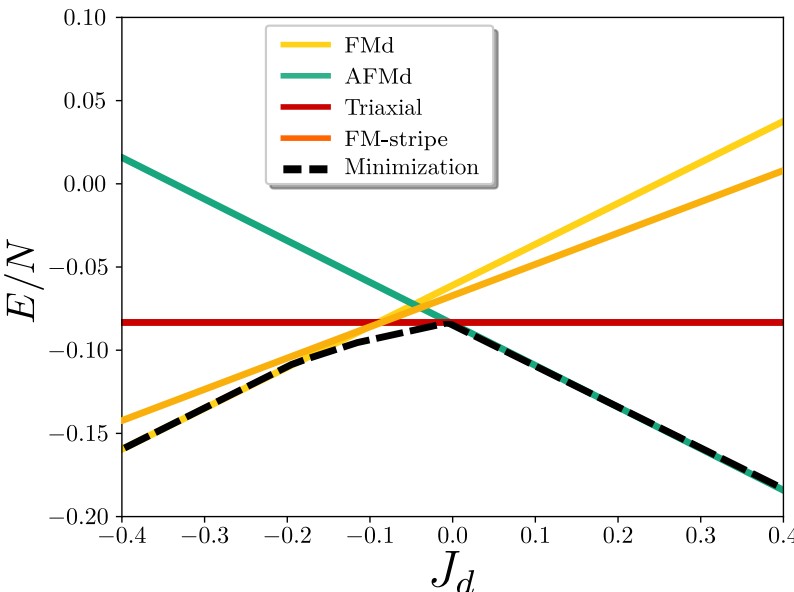

Figure 11: Ground state energy per site for the classically ordered phases: FMd, AFMd and FM-stripe for $J_1 \approx -0.02$ and $J_\chi = 1$. The energy of the triaxial state corresponds to minimizing the chiral term. The energy coming from the gradient descent (GD) minimization, Sec. 3, is also shown for comparison. Same color code as in Fig. 2.

suggest that a massive classical ground state degeneracy persists in the regime of small $J_1$ and $J_d < 0$ up to some critical values beyond which the system enters the FMd or FM-stripe ordered phases.

We can make this argument more quantitative by calculating the classical ground-state energy per site. In Fig. 11 we show the energy for the AFMd, FMd, FM-stripe, and triaxial states—the latter for the pure chiral point—comparing them with the energy of the gradient descent minimization algorithm. In accordance with our qualitative analysis, for $J_d > 0$ the AFMd order is immediately selected out of the set of triaxial states. For $J_d < 0$, on the other hand, the ground state energy remains close to the energy of the triaxial state, and the FMd state is reached only at $J_d \approx -0.15$. Moreover, the energy difference between the FMd and FM-stripe phases is rather small in this region. In the interval $-0.15 \lesssim J_d < 0$, the structure factor displays no Bragg peaks and we interpret this disordered region as partially inheriting the extensive ground state degeneracy of the triaxial state.

## B  Mean-field decoupling of the three-spin interaction

Using the parton representation in Eq. (4), we rewrite the three-spin interaction on a single up-pointing triangle as

$$
\begin{aligned}
H_{ijk} &= J_\chi \mathbf{S}_i \cdot (\mathbf{S}_j \times \mathbf{S}_k) \\
&= \frac{J_\chi}{8} \epsilon_{abc} (\sigma^a)_{\alpha_1\beta_1} (\sigma^b)_{\alpha_2\beta_2} (\sigma^c)_{\alpha_3\beta_3} f^\dagger_{i\alpha_1} f_{i\beta_1} f^\dagger_{j\alpha_2} f_{j\beta_2} f^\dagger_{k\alpha_3} f_{k\beta_3} \\
&= \frac{iJ_\chi}{4} \left[ f^\dagger_{i\uparrow} f_{i\uparrow} f^\dagger_{j\uparrow} f_{j\downarrow} f^\dagger_{k\downarrow} f_{k\uparrow} - f^\dagger_{i\uparrow} f_{i\uparrow} f^\dagger_{j\downarrow} f_{j\uparrow} f^\dagger_{k\uparrow} f_{k\downarrow} + (\text{cyclic perm. } ijk) + (\uparrow \leftrightarrow \downarrow) \right] \\
&= \frac{iJ_\chi}{4} \left[ f^\dagger_{i\uparrow} f_{k\uparrow} f^\dagger_{k\downarrow} f_{j\downarrow} f^\dagger_{j\uparrow} f_{i\uparrow} - f^\dagger_{i\uparrow} f_{j\uparrow} f^\dagger_{j\downarrow} f_{k\downarrow} f^\dagger_{k\uparrow} f_{i\uparrow} + (\text{cyclic perm. } ijk) + (\uparrow \leftrightarrow \downarrow) \right] . \quad (19)
\end{aligned}
$$

There are in total 12 terms in the last line of Eq. (19). In the mean-field approximation, we take $f_{i\uparrow}^\dagger f_{j\uparrow} = \frac{1}{2}\xi_{ij} + \hat{\delta}_{ij,\uparrow}$, where $\hat{\delta}_{ij,\uparrow}$ is the fluctuation. Thus,

$$
\begin{aligned}
H_{ijk} &= \frac{iJ_\chi}{4}\left[\left(\frac{\xi_{ik}}{2}+\hat{\delta}_{ik,\uparrow}\right)\left(\frac{\xi_{kj}}{2}+\hat{\delta}_{kj,\downarrow}\right)\left(\frac{\xi_{ji}}{2}+\hat{\delta}_{ji,\uparrow}\right)\right. \\
&\quad \left. -\left(\frac{\xi_{ij}}{2}+\hat{\delta}_{ij,\uparrow}\right)\left(\frac{\xi_{jk}}{2}+\hat{\delta}_{jk,\downarrow}\right)\left(\frac{\xi_{ki}}{2}+\hat{\delta}_{ki,\uparrow}\right)+(\text{cyclic perm. } ijk)+(\uparrow\leftrightarrow\downarrow)\right] \\
&\approx \frac{iJ_\chi}{16}\left[-\xi_{ik}\xi_{kj}\xi_{ji}+\xi_{kj}\xi_{ji}f_{i\uparrow}^\dagger f_{k\uparrow}+\xi_{ik}\xi_{kj}f_{j\uparrow}^\dagger f_{i\uparrow}+\xi_{ji}\xi_{ik}f_{k\downarrow}^\dagger f_{j\downarrow}+\xi_{ki}\xi_{jk}\xi_{ij}\right. \\
&\quad \left. -\xi_{jk}\xi_{ij}f_{k\uparrow}^\dagger f_{j\uparrow}-\xi_{ki}\xi_{jk}f_{i\uparrow}^\dagger f_{j\uparrow}-\xi_{ij}\xi_{ki}f_{j\downarrow}^\dagger f_{k\downarrow}+(\text{cyclic perm. } ijk)+(\uparrow\leftrightarrow\downarrow)\right] \\
&= \frac{3iJ_\chi}{16}\sum_\alpha\left(-\xi_{ik}\xi_{kj}\xi_{ji}+\xi_{kj}\xi_{ji}f_{i\alpha}^\dagger f_{k\alpha}+\xi_{ik}\xi_{kj}f_{j\alpha}^\dagger f_{i\alpha}+\xi_{ji}\xi_{ik}f_{k\alpha}^\dagger f_{j\alpha}-\text{h.c.}\right). \quad (20)
\end{aligned}
$$

If $ijk$ are oriented counterclockwise as in Fig. 5, we substitute $\xi_{ik}=\xi_{kj}=\xi_{ji}=-i\xi_1$ and obtain

$$
H_{ijk}=\frac{3J_\chi\xi_1^3}{4}-\frac{3iJ_\chi\xi_1^2}{16}\sum_\alpha\left(f_{i\alpha}^\dagger f_{k\alpha}+f_{j\alpha}^\dagger f_{i\alpha}+f_{k\alpha}^\dagger f_{j\alpha}-\text{h.c.}\right). \quad (21)
$$

We also have to take into account the down-pointing triangles, which contribute with hopping outside the unit cell. To count the energy per site, we note that each site belongs to two triangles and the energy of each triangle has to be divided by three sites. At the end, the total mean-field Hamiltonian has the form given in Eqs. (12) and (13).

# Acknowledgements

The authors are thankful to Vitor Dantas for helpful discussions.

**Author contributions**    F.O. and J.A.S. contributed equally to this work.

**Funding information**    We acknowledge funding by Brazilian agencies FAPESP (E.C.A.) and CNPq (F.O., E.C.A., R.G.P.). J.A.S. acknowledges funding by CAPES - Finance Code 001, via Grant No. 88887.474253/2020-00. Research at IIP-UFRN is supported by Brazilian ministries MEC and MCTI. This work was also supported by a grant from Associação Instituto Internacional de Física (R.G.P.).

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
