# Peer review of "Noncoplanar magnetic orders and gapless chiral spin liquid on the kagome lattice with staggered scalar spin chirality"

_SciPost Physics, doi:SciPost Phys. 13, 050 (2022)_

## Round 2 · Referee Report · Anonymous (Referee 1) · 2022-4-22

Strengths

1) The problem considered in his article is interesting, 2) The tools chosen to handle it are appropriate, 3) The article is well-written

Weaknesses

1) Many points need to be corrected and precised (the list is given below). 2) The problem has already been studied before, at the special point with only chiral interactions. The possibility of a chiral spin liquid was already well established. Majorana fermions have been used previously. Is the use of Abrikosov fermions new ? If yes, it should be emphasized. 2) To meet the criteria of publication in Scipost, the challenge must appear more clearly (maybe the question of the stability of the CSL with respect to magnetic order, that had not been explored previously ?)

Report

The authors study a kagome spin model, where the dominant interaction are chiral 3-spin interactions J_chi. The J_chi model has been studied in detail previously. But here, the authors add first neighbor (J1) and one type of third neighbor (Jd) interactions, interpolating between the theoretical model and the Kapellasite compound. They numerically look for the classical ground state and draw the classical phase diagram, with interesting new magnetic orders (AFMd, FMd, FMstr), enclosing a disordered region.
Then, they switch to a quantum model of free fermions, with link parameters on first and third neighbors chosen to describe a chiral spin liquid, and optimize the variational energy of the Gutwiller projection. They also use a parameter h that allows the state to magnetically order. A quantum phase diagram is obtained, where the boundaries have moved with respect to the classical one, and the disordered phase is replaced by a chiral spin liquid one.

The problem considered in his article is interesting, and the tools chosen to handle are appropriate, but many points need to be corrected and precised (the list is given below). Moreover, to meet the criteria of publication in Scipost, the challenge must appear more clearly (maybe the question of the stability of the CSL with respect to magnetic order ?) In the actual state of this article, I cannot accept it in SciPost.

Requested changes

1) In the introduction of Section 3 (classical ground state), could you give the value af the L you use in your simulations ? By the same way, indicate the value used in the snapshots of Fig. 3 ? 2) In Fig. 3, could you use a more contrasted color for the z component of the spin, to make more easy the reading of the spin orientations ? And add the name of the phase ? 3) In Fig. 4, please give the value of Jd that is used. 4) About the point interpolating between cuboc1 and cuboc2, where all Bragg peaks have the same intensity: it seems to me that the state is not the octahedral state of Ref2. In the octahedral state, spins of an horizontal line of Fig.3b through the diagonal of hexagons are ferromagnetic. Here, they are antiferromagnetic. Two different magnetic orders can still have the same Bragg peaks. Is it the case here ? I didn't find the Ref 37 (supplemental material). Could you make it accessible ? 5) As explained in Ref 43, the Heisenberg model, expressed in terms of parton operators, has a SU(2) gauge symmetry. But unlike what is written, the standard parton mean-field decoupling of the Heisenberg interactions has no more this gauge symmetry, but only a U(1) one. 6) Eq. 7 should be explained in a supplemental material, as it was done in Ref 44. 7) After eq. 7, could you precise why you choose this Ansatz (number 11) and not another one of the table IX of Ref. 9 ? Did you try other Anzaetze ? Is it related to the symmetries of the Hamiltonian with the chiral term ? You also tried the number 9. Why this one ? 8) Figure 6, it seems that the titles above plots are false and that kappa_1 and kappa_d have been exchanged (or it is in the text, but in any case, the text and the plots do not correspond). 9) Could you precise what you mean by "mixed boundary conditions" ? Do you mean periodic and antiperiodic boundary conditions in two directions ? 10) When you add h as variational parameter, it implies that the directions of the Si are fixed. For AFMd for example (but it must be the same for FMd), the relative orientations of the chains in the classical ground state depends on the J's. Did you try the orientation of the classical ground state only, or did you also try to optimize the directions of the spins ? 11) Eq. 18, you compute the sublattice magnetization for spins belonging to the same magnetic sublattice. If I understand correctly, it means that you have one type of m for each classical order ? How are these m related to the m of Fig. 4 (the height of the Bragg peaks at the appropriate k) ? 12) Just before section 6, there is a typo in the refereces to Fig 3 (twice 3(e)) 13) In the bibliography, most of capital letters of titles are not capital.

  • validity: high
  • significance: good
  • originality: ok
  • clarity: high
  • formatting: excellent
  • grammar: excellent

Author:  Fabrizio Oliviero  on 2022-05-09  [id 2451]

(in reply to Report 1 on 2022-04-22)
Category:
answer to question
correction

We thank the referee for the very careful reading of our working and for raising important points which helped us improve our manuscript.

Concerning the questions about the novelty and challenge in our work, we would like to clarify that, to our knowledge, the use of Abrikosov fermions to construct a variational wave function for this gapless CSL is new. However, we believe that our main original contribution is to address the question of stability of the gapless CSL with respect to magnetic order, as the referee also pointed out, and to identify the noncoplanar phases that appear around the CSL. Following the referee's remarks, we emphasize the motivation and impact of our work in the new version of our manuscript.

Below we address the list of specific questions and requested changes. The referee writes: "1) In the introduction of Section 3 (classical ground state), could you give the value af the L you use in your simulations ? By the same way, indicate the value used in the snapshots of Fig. 3 ?"

Our response: The snapshots in Fig. 3 were obtained with L=12. The simulations from Section 3 were repeated for L \in [6,20] in order to find possible phases with different magnetic unit cell sizes. This information has been added to the main text and the caption of Fig. 3.

The referee writes: "2) In Fig. 3, could you use a more contrasted color for the z component of the spin, to make more easy the reading of the spin orientations ? And add the name of the phase ?"

Our response: These modifications were adopted in the new version of Fig. 3. The best contrasted color found for the spin configurations (red) is now adopted in Fig. 3.

The referee writes: "3) In Fig. 4, please give the value of Jd that is used."

Our response: The behavior seen in Fig. 4 is the same for all Jd>0 available in the classical phase diagram in Fig. 2. This information is now better explained in the caption of Fig. 4.

The referee writes: "4) About the point interpolating between cuboc1 and cuboc2, where all Bragg peaks have the same intensity: it seems to me that the state is not the octahedral state of Ref2. In the octahedral state, spins of an horizontal line of Fig.3b through the diagonal of hexagons are ferromagnetic. Here, they are antiferromagnetic. Two different magnetic orders can still have the same Bragg peaks. Is it the case here ? I didn't find the Ref 37 (supplemental material). Could you make it accessible ?"

Our response: The referee is correct. That’s the reason why we have indicated “an octahedral state maximizing the staggered chirality emerges” in the caption of Fig. 4. This phase is a slight variation of the octahedral state found in Ref. 2 with AFM chains in the diagonals of the hexagons as noted by the referee. A new sentence explaining this remark has been added to the introduction. The supplemental material is also now available at Ref. 37 (link here: https://github.com/joaosds/suppl_noncoplanar).

The referee writes: "5) As explained in Ref 43, the Heisenberg model, expressed in terms of parton operators, has a SU(2) gauge symmetry. But unlike what is written, the standard parton mean-field decoupling of the Heisenberg interactions has no more this gauge symmetry, but only a U(1) one."

Our response: We agree with the referee. We have corrected the statement about the mean-field Hamiltonian below Eq. 5 accordingly.

The referee writes: "6) Eq. 7 should be explained in a supplemental material, as it was done in Ref 44."

Our response: We have added an appendix to present the detailed derivation of Eq. 7.

The referee writes: "7) After eq. 7, could you precise why you choose this ansatz (number 11) and not another one of the table IX of Ref. 9 ? Did you try other Anzaetze ? Is it related to the symmetries of the Hamiltonian with the chiral term ? You also tried the number 9. Why this one ?"

Our response: The choice of ansatz no. 11 was indeed based on the symmetries of the Hamiltonian, but it was also guided by the spinon spectrum. We focused on No. 11 because it reproduces a spectrum with a line Fermi surface suggested by the previous numerical results of Ref. [26] for the pure-J_chi model. While we have not performed an extensive study of all symmetry-allowed Ansätze classified in Ref. [9], we also considered Ansatz No. 9 as an alternative because it allowed us to investigate whether changing the flux through the trapezoids and allowing for a more general shape of the Fermi surface might lead to a lower-energy state. However, we found that No. 9 was not competitive in terms of the projected ground state energy. We have rewritten the first paragraph of Section 4 to elaborate on our choice of the mean-field ansatz. In particular, we now state explicitly that we have not explored all possible Ansätze.

The referee writes: "8) Figure 6, it seems that the titles above plots are false and that kappa_1 and kappa_d have been exchanged (or it is in the text, but in any case, the text and the plots do not correspond). The text is correct, but the roles of kappa_1 and kappa_d were reversed in the plots."

Our response: We thank the referee for noticing this oversight. We have corrected the plot labels in the new version of our manuscript.

The referee writes: "9) Could you precise what you mean by "mixed boundary conditions" ? Do you mean periodic and antiperiodic boundary conditions in two directions ?"

Our response: Yes, that is correct. This information is now better explained in Sec. 5.

The referee writes: "10) When you add h as variational parameter, it implies that the directions of the Si are fixed. For AFMd for example (but it must be the same for FMd), the relative orientations of the chains in the classical ground state depends on the J's. Did you try the orientation of the classical ground state only, or did you also try to optimize the directions of the spins ?"

Our response: We thank the referee for this remark. The relative orientations of the chains in the AFMd and FMd phases depend on J1, but not on Jd. The referee is correct in saying that one may also optimize this angle within VMC. However, we did not do it here to keep the variational parameters to a minimum, which allowed us to explore the phase diagram in detail. This is the same motivation for not including Jastrow factors in the trial wave function. In short, the classical ground state acts as a rigid staggered field on top of the spin-liquid phase. The sole variational parameter is its strength h. We have added this comment in Sec. 6.

The referee writes: "11) Eq. 18, you compute the sublattice magnetization for spins belonging to the same magnetic sublattice. If I understand correctly, it means that you have one type of m for each classical order ? How are these m related to the m of Fig. 4 (the height of the Bragg peaks at the appropriate k) ?"

Our response: The referee is correct once again. The magnetization from Eq. 18 is physically equivalent to the one studied in the classical approach in Sec. 3. Nevertheless, we note that in the quantum case m < 1/2 due to quantum fluctuations, even though the classical angles are kept fixed, showing that our simple VMC scheme does capture fluctuations inside the ordered phases.

The referee writes: "12) Just before section 6, there is a typo in the references to Fig 3 (twice 3(e))."

Our response: We thank the referee for the careful reading. This is now corrected in the new version of the manuscript.

The referee writes: "13) In the bibliography, most of capital letters of titles are not capital."

Our response: There was a problem with the bibtex file. We have corrected the capital letters in the new version of the manuscript.

List of changes: – – – – – – – – – – – Added a few sentences in the introduction to emphasize the novelty of our results. Added comment about relation to standard octahedral state. Added information about system sizes used in section 3. Changed color of arrows in Fig. 3 and added names of the phases. Added information that Jd>0 to the caption of Fig. 4. Expanded the first paragraph of section 4 to clarify choice of mean-field Ansatz. Changed “SU(2)” to “U(1)” gauge redundancy below Eq. 5. Corrected plot labels in Fig. 6. Added discussion about fixed parameters and boundary conditions used in the VMC in section 5. Added appendix about the derivation of Eq. 7. Corrected a few typos, including capital letters in the bibliography.

---

## Round 2 · Referee Report · Anonymous (Referee 2) · 2022-5-16

Report

In this paper, the authors study the ground-state phase diagram of an extended Heisenberg model on the kagome lattice, including three-site chiral terms.

I have a few remarks that should be addressed:

1) I do not fully understand Fig.3. For example, for the FM state I would expect a single Bragg peak in Gamma (red dot), why are there also other pink dots in the middle of the edge of the Brillouin zone? Similarly, for the AFMd state: why are there both red and pink dots?

Most importantly, I think that the ``disordered'' phase shown in panel (f) is the result of a (possibly large) ground state degeneracy. It would be important to clarify what is the family of degenerate ground states.

2) Within the parton mean-field calculation, only one chiral state is analysed, is there a reason to neglect the other ones considered in table IX of Ref.[9]? Does it correspond to the lowest-energy state (at the mean-field level)? What are the optimal fluxes?

3) It looks to me that very few Monte Carlo steps have been used to compute observables. In this way, errobars could be underestimated. It is also not clear to me if the parameters of the unprojected wave function are optimized within a Monte Carlo procedure (i.e., in presence of the Gutzwiller projection) or at the mean-field level. It is known that the Gutzwiller projection may strongly change the actual values of the fermionic hoppings in Eq.(15), possibly changing the nature of the wave function.

4) Computing the magnetization by using the spin-spin correlation at the largest distance as in Eq.(18) may be dangerous in presence of incommensurate spin orders, where spin-spin correlations may have large oscillations. A consequence of this behavior could be the underestimation of the actual magnetization. In this sense, it would be better to consider the full information contained in the structure factor.

5) Chiral spin liquids have been studied by similar variational wave functions in the Heisenberg model in presence of third-nearest-neighbor super-exchange interactions in Phys. Rev. B 91, 041124 (2015). In this paper, also an explicit three-site chiral term has been considered. This work could be cited.

In summary, I think that the paper could be published, after these points will be addressed.

  • validity: good
  • significance: good
  • originality: good
  • clarity: good
  • formatting: good
  • grammar: good

Author:  Fabrizio Oliviero  on 2022-05-30  [id 2545]

(in reply to Report 2 on 2022-05-16)
Category:
answer to question
pointer to related literature

We thank the referee for the very careful reading of our work and for raising important points which helped us improve our manuscript.

Below we address the list of specific questions and requested changes.

The referee writes:

"1) I do not fully understand Fig.3. For example, for the FM state I would expect a single Bragg peak in Gamma (red dot), why are there also other pink dots in the middle of the edge of the Brillouin zone? Similarly, for the AFMd state: why are there both red and pink dots?"

Our response:

We thank the referee for the remark. The referee is correct in expecting a Bragg peak at the Gamma point for the FM phase. Our reasoning for showing the extra points at the boundary of the first Brillouin zone is to highlight that their weight is not the same due to the 3-site unit cell. That is why they come in different colors.

For the AFMd phase, we follow the same idea: the different colors suggest different magnitudes of the Bragg peaks.

We add the following sentence to the caption of Fig.3 : “The inner (outer) hexagon in the structure factor represents the original (extended) Brillouin Zone of the kagome lattice. In addition, the darker (lighter) the dot, the stronger (weaker) the relative intensity of the Bragg peaks located at this position.”

The referee writes:

"2) Within the parton mean-field calculation, only one chiral state is analysed, is there a reason to neglect the other ones considered in table IX of Ref.[9]? Does it correspond to the lowest-energy state (at the mean-field level)? What are the optimal fluxes?"

Our response:

The choice of ansatz no. 11 in table IX of Ref. [9] was based on the symmetries of the Hamiltonian and guided by the spinon spectrum. We focused on No. 11 because it reproduces a spectrum with a line Fermi surface suggested by the previous numerical results of Ref. [26] for the pure-J_chi model. While we have not performed an extensive study of all symmetry-allowed Ansätze classified in Ref. [9], we also considered Ansatz No. 9 as an alternative because it allowed us to investigate whether changing the flux through the trapezoids might lead to a lower-energy state. However, we found that No. 9 was not competitive in terms of the projected ground state energy, which provides a more precise criterion than the energy at the mean-field level.

In the revised version of our manuscript we elaborate on the reasons behind the choice of the mean-field ansatz.

The referee writes:

"3) It looks to me that very few Monte Carlo steps have been used to compute observables. In this way, errobars could be underestimated. It is also not clear to me if the parameters of the unprojected wave function are optimized within a Monte Carlo procedure (i.e., in presence of the Gutzwiller projection) or at the mean-field level. It is known that the Gutzwiller projection may strongly change the actual values of the fermionic hoppings in Eq.(15), possibly changing the nature of the wave function."

Our response:

We thank the referee for the remark. We carefully checked that the ground state energy is converged in all results, and we have absolute precision of 10^-4 for L > 8. We benchmarked our code against published VMC results for the Heisenberg model in the kagome lattice, Refs. [47-51].

For other observables, the situation is more subtle. For instance, we do not expect the magnetization to converge to the same degree of precision as the energy since the VMC is designed to minimize E. Nevertheless, we obtain good results, which allow us to draw a complete phase diagram of the problem. As we discuss on page 11, the primary source of error comes from the presence of magnetization plateaus, which hampers a precise identification of the phase boundaries. This is similar to what happens in exact diagonalization studies of the Heisenberg model in small clusters. Unfortunately, the current system sizes accessible in VMC simulations are not sufficient to obliviate this problem, even with our carefully chosen boundary conditions to the mean-field solution.

As for the parameter optimization, they are performed after the projection. For instance, Fig. 7 shows the ground state energy after projection as a function of the mean-field parameters. Incidentally, we limit our optimization range to the stability region of the mean-field solution.

The referee writes:

"4) Computing the magnetization by using the spin-spin correlation at the largest distance as in Eq.(18) may be dangerous in presence of incommensurate spin orders, where spin-spin correlations may have large oscillations. A consequence of this behavior could be the underestimation of the actual magnetization. In this sense, it would be better to consider the full information contained in the structure factor. "

Our response:

We thank the referee for this observation. All ordered phases are commensurate in our work, and we do not observe oscillations in the order parameter, similar to what is reported in Ref. [47], for instance.

Calculating the full structure factor would indeed produce a more complete picture of the problem. However, it is also more time-consuming and would not bring new fundamental insights for the commensurate phases we study.

In the new version of the manuscript, we add the following sentence after Eq. 18: “This definition of the order parameter is sufficient to study the commensurate ordered phases identified in this work."

The referee writes:

"5) Chiral spin liquids have been studied by similar variational wave functions in the Heisenberg model in presence of third-nearest-neighbor super-exchange interactions in Phys. Rev. B 91, 041124 (2015). In this paper, also an explicit three-site chiral term has been considered. This work could be cited."

Our response:

The referee is correct and we thank them for the reminder. A new reference to this work will be included in the Introduction for the final version of the manuscript.

List of changes:

Added a new sentence after eq. (18) about the definition of the magnetization.
Added new sentences to the caption of Fig. 3 about the intensity of Bragg peaks and definition of the 1st and extended Brillouin zone.
Added a new reference to Phys. Rev. B 91, 041124 (2015) to the 2nd paragraph of the introduction as recommended by the referee.

---

## Round 3 · Author Response

Thank you for the opportunity to submit our paper once more for the SciPost physics journal. We believe the additional changes have helped us to substantially improve our manuscript. We thank both referees for the constructive comments on our manuscript and the opportunity to answer their questions. Our responses were posted on the submission webpage and are viewable online.
Sincerely,
The Authors.

---

## Round 3 · List of Changes

List of changes suggested by the first referee:
• Added a few sentences in the introduction to emphasize the novelty of our results.
• Added comment about relation to standard octahedral state.
• Added information about system sizes used in section 3.
• Changed color of arrows in Fig. 3 and added names of the phases.
• Added information that Jd>0 to the caption of Fig. 4.
• Expanded the first paragraph of section 4 to clarify choice of mean-field Ansatz.
• Changed “SU(2)” to “U(1)” gauge redundancy below Eq. 5.
• Corrected plot labels in Fig. 6.
• Added discussion about fixed parameters and boundary conditions used in the VMC in section 5.
• Added appendix about the derivation of Eq. 7.
• Corrected a few typos, including capital letters in the bibliography.
List of changes suggested by the second referee
• Added a new sentence after eq. (18) about the definition of the magnetization.
• Added new sentences to the caption of Fig. 3 about the intensity of Bragg peaks and definition of the 1st and extended Brillouin zone.
• Added a new reference to Phys. Rev. B 91, 041124 (2015) to the 2nd paragraph of the introduction as recommended by the referee.

---

## Round 4 · Author Response

Dear Editor in charge,

Thank you for the opportunity to submit our paper once more for the SciPost physics journal. As requested by the EIC, we did a minor revision of the manuscript. We believe that the current version of the manuscript was improved by the additional changes suggested by the referees. Once again, we thank both referees for the constructive comments on our work and the chance to answer their questions.

Sincerely,

The Authors.

---

## Round 4 · List of Changes

List of changes:

1- We added an appendix which explains the origin of the disordered region in the classical phase diagram showed in section 3. 2- Added new reference Ref. (55).

---

## Editorial Decision

published